# Adhesion of Flowable Resin Composites in Simulated Wedge-Shaped Cervical Lesions: An In Vitro Pilot Study



Diana Bănuț Oneț [1,†], Lucian Barbu Tudoran [2,3,†], Ada Gabriela Delean [4,†], Petra Șurlin [5,†], Andreea Ciurea [1,†], Alexandra Roman [1,*,†], Sorana D. Bolboacă [6], Cristina Gasparik [7,†], Alexandrina Muntean [8,†] and Andrada Soancă [1,†]

1. Department of Periodontology, Faculty of Dental Medicine, Iuliu Hațieganu University of Medicine and Pharmacy Cluj-Napoca, Victor Babeş St., No. 15, 400012 Cluj-Napoca, Romania; dyabanut@yahoo.ro (D.B.O.); andreea_candea@yahoo.com (A.C.); andrapopovici@gmail.com (A.S.)
2. Electron Microscopy Center, Department of Molecular Biology and Biotechnologies, Faculty of Biology and Geology, Babeş-Bolyai University, Clinicilor St., No. 5-7, 400006 Cluj-Napoca, Romania; lucianbarbu@yahoo.com
3. Electron Microscopy Integrated Laboratory (LIME), National Institute for Research and Development of Isotopic and Molecular Technologies, INCDTIM, 67-103 Donath St., 400293 Cluj-Napoca, Romania
4. Department of Odontology and Endodontics, Faculty of Dental Medicine, Iuliu Hațieganu University of Medicine and Pharmacy Cluj-Napoca, Motilor St., No. 33, 400012 Cluj-Napoca, Romania; ada.delean@umfcluj.ro
5. Department of Periodontology, Faculty of Dental Medicine, University of Medicine and Pharmacy of Craiova, Petru Rareș St., No. 2, 200349 Craiova, Romania; surlinpetra@gmail.com
6. Department of Medical Informatics and Biostatistics, Iuliu Hațieganu University of Medicine and Pharmacy Cluj-Napoca, Louis Pasteur St., No. 6, 400349 Cluj-Napoca, Romania; sbolboaca@gmail.com
7. Division Dental Propaedeutics, Aesthetics, Department of Prosthetic Dentistry and Dental Materials, Faculty of Dental Medicine, Iuliu Hatieganu University of Medicine and Pharmacy Cluj-Napoca, Clinicilor St., No. 31, 400001 Cluj-Napoca, Romania; gasparik.cristina@umfcluj.ro
8. Department of Paediatric Dentistry, Faculty of Dental Medicine, Iuliu Hațieganu University of Medicine and Pharmacy Cluj-Napoca, Avram Iancu St., No. 31, 400083 Cluj-Napoca, Romania; ortoanda@yahoo.com
* Correspondence: veve_alexandra@yahoo.com; Tel.: +40-264-597256
† These authors contributed equally to the present study and should be regarded as main authors.

**Abstract:** The resin composite restoration of non-carious cervical lesions (NCCLs) still faces some drawbacks mostly related to the quality of the marginal seal. This study attempts to evaluate the adhesive capacities of two flowable and two conventional hybrid resin composite restorations of NCCLs having two types of cervical margins. Our null hypothesis assumes the same adhesive behavior of different materials. The relative composition of dental–restoration structures was also measured. Thus, restored wedge-shaped cervical cavities were realized on both the buccal and oral surfaces of extracted teeth. After immersion in dye solution, sectioning of the teeth was performed. We proposed an optical microscopy method to quantify the dye penetration along the restoration–tooth interface and scanning electron microscopy (SEM) and energy-dispersive X-ray analysis (EDX) to evaluate the quality of the peripheral seal. The data obtained revealed an amount of dentinal microleakage for all tested materials, despite the favorable results of the restoration peripheral seal. Therefore, data from this study failed to reject the null hypothesis. The adhesion is not influenced by the position of cervical margins. The SEM revealed occasional disruptions of the adhesive interface. EDX sustains the qualitative compositions as provided by the manufacturers. Conclusions: The four experimental composites are recommended to restore NCCLs in clinic.

**Keywords:** restorative materials; properties; adhesion; microleakage; scanning electron microscopy

## 1. Introduction

Non-carious cervical lesions (NCCLs) are defined as a loss of dental hard tissues at the cemento-enamel junction area through processes unrelated to caries [1]. Nowadays,



they are a more and more common pathology in clinical practice [2]. The development of NCCLs is a consequence of changes in lifestyle such as diet and oral hygiene habits as well as of increasing life expectancy [3]. NCCLs are usually produced by a combination of abrasive, erosive, and stress-producing factors, which generate different clinical appearances varying from shallow lesions with poorly defined margins to large wedge-shaped defects [1]. NCCLs are almost exclusively situated on the facial surfaces of teeth [1]. The identification and the management of the putative etiologic factors are mandatory before initiating treatment [4]. To date, no conclusive evidence for reliable, predictable, and successful treatment strategies for NCCLs exists [1]. Depending on the clinical appearance and the localization of NCCL (i.e., coronal or radicular), the current treatment options include monitoring and prevention [4], treatment of dentin hypersensitivity [1], restorative treatment [2], or root coverage surgery associated or not with restorative treatment [5].

If ignored, NCCLs invariably enlarge [2]. Although monitoring NCCL could be a clinical approach for incipient lesions, the placement of adhesive restorations should be considered for larger crown-located defects. Composite resins combined with adhesive systems are the preferred choice to restore NCCLs in clinical practice [6,7].

The clinical performance of NCCL restorations is highly dependent on the adhesive system used and on the quality of the clinical restorative procedure [1,6,8]. However, other factors have been reported to influence the performance of NCCL restorations, such as the marginal location in enamel or cementum, the quality of the dentin substrate [9,10], the restorative materials [10], and the cavity geometry [11,12]. More recent information reported that only tooth location or the presence of wear facets play an important role in the fate of NCCL restorations [13].

The marginal degradation of composites is frequently seen during the aging of the restorations [1]. Still, primary marginal imperfection consecutive to the localized failures of the adhesion phenomenon is a current weakness reported in cervical restorations, which impacts the longevity and the fate of NCCL restorations [14]. Secondary decay, gingival inflammation due to plaque accumulation [15], and debonding and restoration loss are clinical consequences due to the primary adhesion deficiencies [14]. To protect the adhesive interface from cervical flexural stress, NCCLs with an abfraction component should be restored with a microfilled resin composite or a flowable resin that has a low modulus of elasticity, which allows a discrete flexion in concordance with tooth movement. There is evasive information on the differences between failure rates of resin composites of different stiffness when restoring NCCLs [9,16].

The clinical applications of flowable composites have been partially restricted principally by their mechanical shortcomings [17]. On the other hand, the increased proportion of diluent monomers in flowable composites induces higher polymerization shrinkage and, consequently, an augmented stress at the adhesive interface [18]. Flowable composites have an average polymerization shrinkage rate of 5% [17], while this rate of conventional composites could be less than 1% [19]. New nanotechnology approaches have allowed the development of flowable composites with improved mechanical and esthetic properties and minimized polymerization shrinkage by almost 20% [20]. Flowable composites seem to be a rational approach to restore NCCLs due to their lower modulus of elasticity as compared to packable composites, which could absorb the polymerization shrinkage and flexural stress [7]. The results of clinical studies have not reported an improved retention rate of flowable composites compared with conventional ones [21,22]. Inconsistent information exists in relation to factors influencing the fate of NCCL restorations [10,11,13] and the clinical retention rate of flowable composites in comparison with conventional composites. Future research is encouraged to collect information on these issues [13].

Conventional investigations, such as dye penetration tests, are performed to evaluate resin composites' adhesive properties [23]. Supplemental experiments need to be conducted in order to assess the specificities of the adhesion process and to examine the properties of the material structure. Thus, the use of scanning electron microscopy (SEM) would allow for an in-depth examination of the adhesive interface and tooth structures [24,25], whereas

a semi-quantitative assessment of the chemical elements of substrates could principally be obtained using energy dispersive X-ray analysis (EDX) [26]. It is worth mentioning that EDX studies have found less application in dentistry [26–28], being especially associated with the fields of engineering [29] and chemistry [30].

By coupling the method of optical microscopy with SEM, the aim of this in vitro study is to evaluate the adhesive properties of flowable resin composites placed to restore NCCLs compared with conventional hybrid composite resins. As previously specified, our null hypothesis is based on the assumption that the adhesive behaviors of distinct resin composites are not different. The influence of the cervical substrate on the adhesion performance was also investigated. In addition, EDX analyses were also carried out to evaluate the relative composition of the tooth–restoration structures.

## 2. Materials and Methods

### 2.1. Study Design

A pilot study was conducted on extracted teeth for orthodontic and periodontal problems. Patients were asked to provide informed consent for tooth extraction. In addition, the protocol implemented for our research received approval from the Ethical Board of the Iuliu Haţieganu University of Medicine and Pharmacy Cluj-Napoca (No. 268/30.07.2019). This study was also carried out in compliance with all relevant guidelines and regulations regarding tooth collection.

Four groups of five carie-free premolars each (total of 20 teeth) (Figure S1a) were used in this experiment.

A summary of the research development is illustrated in Figure 1. Briefly, on each tooth, two cervical cavities were prepared, one located on the buccal surface and the other located on the oral surface. Four types of resin-based restorative materials (two conventional composites and two flowable ones) were used to restore the four groups of teeth, using the protocols recommended by manufacturers. Briefly, the restoring steps included selective enamel etching, one-step self-etch adhesive application, and two-layer composite resin applications. The restored teeth were isolated with nail varnish, immersed in dye solution, and then sectioned with a diamond saw, resulting in four sections per tooth, as described before [26]. Two middle sections of each tooth were observed on an optical microscope for quantifying the eventual infiltrations (adhesion failure). Then, the sections were prepared and observed at SEM and investigated by EDX.

### 2.2. Resin-Based Restorative Materials and Adhesive Systems

Cervical cavities were filled with two conventional and two flowable resin composites, shade A3. The detailed composition of the materials and the adhesive systems is provided in Table 1.

The B and BF materials contain Shofu's Giomer technology [19], which is a bioactive surface pre-reacted glass (S-PRG) filler that actively releases six beneficial ions, including fluoride.

### 2.3. Cavity Restoration and Sample Preparation

The extracted teeth were cleaned with an ultrasonic scaler (Suprasson® P5 Booster, SATELEC ACTEON Group, Merignac, France) to eliminate any soft and hard deposits. In addition, 4% chloramine-T storage media was used, and the duration of storage for the extracted teeth was limited to thirty days after the extraction procedure [31].

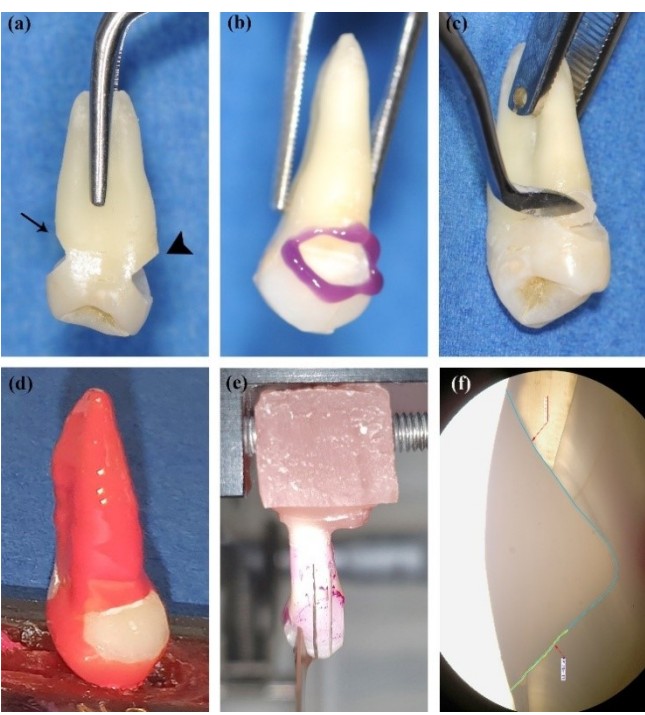

**Figure 1.** Experimental flow for sections preparation. Two prepared cervical cavities with cervical margin in enamel (arrow) and in cementum (arrowhead), respectively (**a**). Selective etching of enamel margins (**b**). Restoration with Beautifil®: application of the gingival layer (**c**). Restored tooth coated with nail varnish (**d**). Tooth sectioned in a buccal–oral direction (**e**). Restoration and dental tissues on optical microscopy. The green line is the length (μm) of cervical microleakage and the blue one is the length of the total adhesion interface (40×) (**f**).

**Table 1.** Composition of the materials and of the adhesive system used in the study.

| Type of Material | Restorative Material/Abbreviation | Manufacturer | Matrix Monomers | Filler Content | Adhesive System |
|---|---|---|---|---|---|
| Low shrinkage nano-hybrid composite resin | Beautifil II® LS (B) | Shofu Dental Corporation, JAPAN | Bis-GMA TEGDMA UDMA Bis-MPEPP | 83% weight S-PRG filler based on Fluoroboroalumino–silicate glass mean particle size 0.4 μm Polymerization initiator, Pigments and others | BeautiBond (Shofu, Japan) "All-in-One" 7th Generation -pH 2.4 -HEMA-free -Bis-GMA -TEGDMA -Phosphonic and carboxylic acid monomers -Acetone -Distilled water |
| Low flow nano-hybrid composite resin | Beautifil Flow Plus® F03 (BF) | Shofu Dental Corporation, JAPAN | Bis-GMA TEGDMA | 67% weight S-PRG filler based on Fluoroboroalumino–silicate glass Polymerization initiator, pigments and others | |
| Microhybrid composite resin | Dynamic Plus® (D) | President Dental, GERMANY | Bis-GMA TEGDMA | 80% weight Barium aluminosilicate—mean particle size ≤ 1 μm Fumed silica—mean particle size ≤ 0.04 μm | Prebond SE (President Dental, Germany) "All-in-One" 7th Generation -pH 3 -functional MDP monomers -4-META -HEMA -Ethanol -Isomer\Bis-GMA -TEGDMA -Aliphatic UDMA -Photoinitiators, water |
| Microhybrid composite | Dynamic Flow (DF) | President Dental, GERMANY | BIS-GMA UDMA. Bis-EMA TMPTMA | Barium aluminum borosilicate <60% | |

LS = Low Shrinkage, S-PRG Technology = surface pre-reacted glass ionomer, Bis-GMA = bisphenol-A-diglycidyl methacrylate; TEGDMA = triethyleneglycol dimethacrylate; UDMA = urethane dimethacrylate; Bis-MPEPP = bisphenol A polyethoxy dimethacrylate, HEMA = 2-hydroxylethyl methacrylate; MDP monomer = 10-methacryloyloxydecyl dihydrogen phosphate; TMPTMA = trimethylol-propane trimethacrylate; Bis-EMA = ethoxylate bisphenol A dimethacrylate: a-META = 4-methacryloxyethyl trimellitic anhydride.

Wedge-shaped cavities resembling the clinical situations were done on the buccal and oral surfaces. The cavities had a mesiodistal width of 3 mm, occlusal–gingival height of 3 mm, and axial depth of 1.5 mm. For one cervical cavity, the gingival margin was located 1 mm apically to the cement–enamel junction (CEJ) (cementum-located cavity), and for the opposite cavity, the gingival margin was located in enamel, just coronally to the CEJ (enamel-located cavity) (Figure 1a). An air, water cooling-high-speed handpiece and diamond-coated burs (Komet S6801.314.014; Komet, Lemgo, Germany) were utilized to prepare the cavities. Discarding of burs was performed after each five-cavity preparation. After the preparation of the wedge-shaped lesions, the teeth were kept in distilled water following former protocols [26,32].

One type of resin composite and the associated adhesive system restored one set of teeth in compliance with the manufacturer's guidelines. A selective enamel etching with 36% phosphoric acid (Blue Etch®, Cerkamed, Stalowa Wola, Poland) was done (Figure 1b). The application of composite materials started from the gingival margin using a two-layer technique and 20 s conventional light-curing (Demetron A2 light-curing unit, Kerr Corporation, Middleton, WI, USA, wavelength 450–470 nm, and light intensity of 1000 mW/cm$^2$) for each increment (Figure 1c). Once the restoration was completed and polished with medium and fine polishing discs (Optidisc™, Kerrhawe SA, Bioggio, Switzerland), teeth were immersed in distilled water at 37 °C and stored until use.

### 2.4. Microleakage Test

Utility wax was used to seal the root apices of the experimental filled teeth. The surfaces of restored teeth were covered with two layers of nail varnish. About 1 mm width around the restoration margins were kept uncoated (Figure 1d). The teeth were introduced in 0.5% basic fuchsine solution for 24 h and then abundantly washed in water for about 10 min. The teeth were embedded in autopolymerizing acrylic resin (Duracryl® Plus, SpofaDental, Jicin, Czech Republic) using a silicon mounting template (Zetaplus & Indurent gel, Zhermack, Badia Polesine, Italy) with the experimental surfaces of the tooth exposed (Figure S1b,c).

Each tooth was sectioned in the buccal–oral direction using a low-speed diamond saw (Isomet, Buehler Ltd., Lake Bluff, IL, USA), resulting in two middle sections (sections) of 1 mm width per tooth used for evaluation (Figure 1e). Thus, ten sections per group of teeth with two zones of interest per section representing the two restorations were obtained. A total of 80 zones of interest (half sections) were analyzed.

The penetrated dye at the level of the dental–restoration interface was appreciated with an inverted microscope (Olympus KC301, Olympus America Inc., Los Angeles, CA, USA) at 40× magnification. Gingival and occlusal microleakage dimensions were measured using QuickPhoto Micro 2.2 software (Olympus, Inc.). The quantification of the microleakage at the dental–restoration interface was done by applying a former approach elaborated by our investigation group [26,32,33] based on a variant of a gap evaluating methods previously reported [32].

Microleakage lengths (in μm) were expressed in terms of value of dye infiltration in relation to the value of the dental-material interface. The length of microleakage and total adhesive interface were provided and inserted by the software on optical microscopic images (Figure 1f). The microleakage proportion was calculated by the ratio (% μm) of the dye penetration length and the tooth–restoration interface length. The ratios of marginal microleakage at the cervical and occlusal levels of the restorations (RMLKc and RMLKo) were calculated for both types of cavities. The absence of microleakage relevant to a good adhesive phenomenon was recorded as a value of zero.

### 2.5. Scanning Electron Microscope (SEM) and Energy-Dispersive X-ray (EDX) Analyses

The sections were prepared for SEM and EDX observations using the same specimens for both approaches. The sections were dried in a graded ethanol series (50–100%) and then sputter-coated with 7 nm gold (Agar Automated Sputter-coater, Agar Scientific Ltd, Essex,

UK). SEM imaging was performed at 30 kV under high-vacuum conditions (Hitachi SU8230 STEM, Hitachi High-Technologies Corporation, Tokyo, Japan). Two photomicrographs were recorded for each sample to thoroughly describe the tooth–restoration interface (the quality of dentin-resinous layer adaptation, the hybrid layer) and the material's structures. Images were analyzed by a single examiner (LBT) using a blinded protocol and measurements of the hybrid layers thickness and the lengths of resinous tags invagination into the dentin were done using ImageJ software. EDX spectra were acquired under the following conditions: 10 μA extraction current, 15 mm working distance, an Oxford Instruments EDS (X-Max$^N$ 80) detector placed inside the sample chamber, and AZtec Software (Oxford Instruments, High Wycombe, UK). Measurements were done at the same magnifications of similar surfaces (three regions of interest - ROIs) for all samples: the healthy dental structures, the resin composite material, and the transition material between the healthy dental structure and the resin composite. Ten measurements for each material were averaged to provide a single mean value for each parameter for each specimen.

### 2.6. Data Analysis

Centrality and dispersion metrics were used to report the microleakage, adhesive lengths, and the ratio of dentin microleakage length (adhesive interface length). The dentin microleakage lengths (μm) ratios were compared using Kruskal–Wallis ANOVA among materials as independent groups at an adjusted significance level of 1.25%. The comparison of the parameters between enamel-located and cementum-located restoration groups was made using Wilcoxon test at a significance level of 5%. For EDX analysis, the average concentrations of elements and standard deviations were calculated for each material.

## 3. Results

### 3.1. Microleakage Test

Wedge-shaped cervical cavities prepared on both buccal and oral surfaces of extracted premolars were restored with two hybrid conventional (Beautiful II LS$^®$/B, Shofu Dental Corporation, Kyoto, Japan; Dynamic Plus$^®$/D, President Dental, Munchen, Germany) and two flowable (Beautiful Flow Plus$^®$ F03/BF, Shofu Dental Corporation, Kyoto, Japan; Dynamic Flow$^®$/DF, President Dental, Munchen, Germany) resin composites. On each tooth, the cervical borders of the two cavities were located in enamel and cementum, respectively. After being removed from dye solution, sectioning of the teeth was accomplished followed by an evaluation of the dye penetration at the restoration-tooth interface. Subsequently, optical microscopy was employed to quantify the marginal microleakage at the restoration site. A combination of SEM and EDX was considered an optimal method to observe the quality of the adhesive interface and its neighboring structures. The relative compositions of restoration–tooth structures were also provided.

The cervical and occlusal marginal microleakage ratios (RMLKc and RMLKo) were calculated for both types of cavities using optical microscopy images. Therefore, Table 2 provides an overview of the lengths of dye penetration measured for the sections of tested materials and their ratios related to the length of the adhesive interface for enamel-located restorations. The data for cementum-located restorations are shown in Table 3.

No significant association has been identified between the type of material and the efficacy (zero infiltration observed for 7/10 B sections, 6/10 BF sections, 4/10 D sections, 6/10 DF sections) for RMLKc in enamel-located restoration (Fisher exact test: *p*-value = 0.60323). Excepting the D where the efficacy was 8/10 sections, all other materials showed the efficacy of 10/10 sections for RMLKo, but the difference does not reach the statistical significance (Fisher exact test: *p*-value = 0.20193).

No significant association has been identified between the type of material and the efficacy (zero infiltration observed for 8/10 B sections, 8/10 BF sections, 3/10 D sections, 6/10 DF sections) for RMLKc in cementum-located restorations (Fisher exact test: *p*-value = 0.06189). Except for the D (9/10 sections) and DF (8/10 sections), all other mate-

rials showed the efficacy of 10/10 for RMLKo, without statistically significant association (Fisher exact test: *p*-value = 0.32186).

No significant differences between enamel-located and cementum-located restorations were observed regarding RMLKc (Figure 2(a1,a2), *p* = 0.3570) or RMLKo (Figure 2(b1,b2), *p* = 0.502).

**Table 2.** Data summary collected for microleakage test based on material type: enamel-located restorations.

| Material | MLKc | MLKo | LAI | RMLKc | RMLKo |
|---|---|---|---|---|---|
| B (n = 10) | | | | | |
| {Min to Max} | {0 to 431} | {0 to 0} | {1546 to 2138} | {0 to 0.28} | {0 to 0} |
| Median (Q1 to Q3) | 0 (0 to 187.5) | 0 (0 to 0) | 1948 (1751.3 to 2028.5) | 0 (0 to 0.11) | 0 (0 to 0) |
| Mean (SD) | 96.4 (161.7) | 0 (0) | 1894.2 (190.8) | 0.06 (0.1) | 0 (0) |
| BF (n = 10) | | | | | |
| {Min to Max} | {0 to 1347} | {0 to 0} | {1252 to 2010} | {0 to 0.72} | {0 to 0} |
| Median (Q1 to Q3) | 0 (0 to 417.3) | 0 (0 to 0) | 1818 (1630 to 1927.5) | 0 (0 to 0.21) | 0 (0 to 0) |
| Mean (SD) | 301.5 (474.4) | 0 (0) | 1752.3 (235.1) | 0.18 (0.29) | 0 (0) |
| D (n = 10) | | | | | |
| {Min to Max} | {0 to 1856} | {0 to 300} | {1788 to 2007} | {0 to 1} | {0 to 0.16} |
| Median (Q1 to Q3) | 470 (0 to 1083.3) | 0 (0 to 0) | 1862 (1852.3 to 1887.3) | 0.24 (0 to 0.57) | 0 (0 to 0) |
| Mean (SD) | 626.9 (675.7) | 40.2 (96.7) | 1875.8 (60) | 0.33 (0.36) | 0.02 (0.05) |
| DF (n = 10) | | | | | |
| {Min to Max} | {0 to 1364} | {0 to 0} | {1576 to 2040} | {0 to 0.72} | {0 to 0} |
| Median (Q1 to Q3) | 0 (0 to 102.3) | 0 (0 to 0) | 1884.5 (1820.5 to 1945.8) | 0 (0 to 0.05) | 0 (0 to 0) |
| Mean (SD) | 279.2 (538.6) | 0 (0) | 1862.6 (134.2) | 0.15 (0.29) | 0 (0) |
| *p*-value * | 0.2488 | 0.1044 | 0.4997 | 0.3128 | 0.1044 |

MLKc = cervical marginal microleakage; MLKo = occlusal marginal microleakage; LAI = length of adhesive interface; RMLKc = ratio of cervical marginal microleakage; RMLKo = ratio of occlusal marginal microleakage; B = Beautiful II LS®; BF = Beautiful Flow Plus® F03; D = Dynamic Plus®; DF = Dynamic Flow®; Q1 = 25th percentile; Q3 = 75th percentile; SD = standard deviation. * Kruskal-Wallis test.

**Table 3.** Data summary collected for microleakage test based on material type: cementum-located restorations.

| Material | MLKc | MLKo | LAI | RMLKc | RMLKo |
|---|---|---|---|---|---|
| B (n = 10) | | | | | |
| {Min to Max} | {0 to 784} | {0 to 0} | {1527 to 2017} | {0 to 0.41} | {0 to 0} |
| Median (Q1 to Q3) | 0 (0 to 0) | 0 (0 to 0) | 1875 (1755.25 to 1965.25) | 0 (0 to 0) | 0 (0 to 0) |
| Mean (SD) | 129 (279.7) | 0 (0) | 1845.4 (152.9) | 0.07 (0.14) | 0 (0) |
| BF (n = 10) | | | | | |
| {Min to Max} | {0 to 1440} | {0 to 0} | {1529 to 2081} | {0 to 0.75} | {0 to 0} |
| Median (Q1 to Q3) | 0 (0 to 0) | 0 (0 to 0) | 1881.5 (1743.5 to 2026.8) | 0 (0 to 0) | 0 (0 to 0) |
| Mean (SD) | 188 (461.1) | 0 (0) | 1857.1 (199.3) | 0.1 (0.24) | 0 (0) |
| D (n = 10) | | | | | |
| {Min to Max} | {0 to 2291} | {0 to 898} | {1504 to 2319} | {0 to 1} | {0 to 0.39} |
| Median (Q1 to Q3) | 366.5 (10.5 to 1193.8) | 0 (0 to 0) | 1912.5 (1822 to 2128.3) | 0.16 (0 to 0.65) | 0 (0 to 0) |
| Mean (SD) | 666.5 (788) | 89.8 (284) | 1957.9 (257.9) | 0.35 (0.4) | 0.04 (0.12) |
| DF (n = 10) | | | | | |
| {Min to Max} | {0 to 998} | {0 to 645} | {1231 to 2278} | {0 to 0.59} | {0 to 0.3} |
| Median (Q1 to Q3) | 0 (0 to 197) | 0 (0 to 0) | 1942.5 (1701.5 to 2149.8) | 0 (0 to 0.09) | 0 (0 to 0) |
| Mean (SD) | 157 (311.7) | 71.5 (202.7) | 1876.9 (330.5) | 0.09 (0.18) | 0.03 (0.09) |
| *p*-value * | 0.0713 | 0.2928 | 0.7689 | 0.0653 | 0.2928 |

MLKc = cervical marginal microleakage MLKo = occlusal marginal microleakage; LAI = length of adhesive interface; RMLKc = ratio of cervical marginal microleakage; RMLKo = ratio of occlusal marginal microleakage; B = Beautiful II LS®; BF = Beautiful Flow Plus® F03; D = Dynamic Plus®; DF = Dynamic Flow®; Q1 = 25th percentile; Q3 = 75th percentile; SD = standard deviations; * Kruskal-Wallis test.

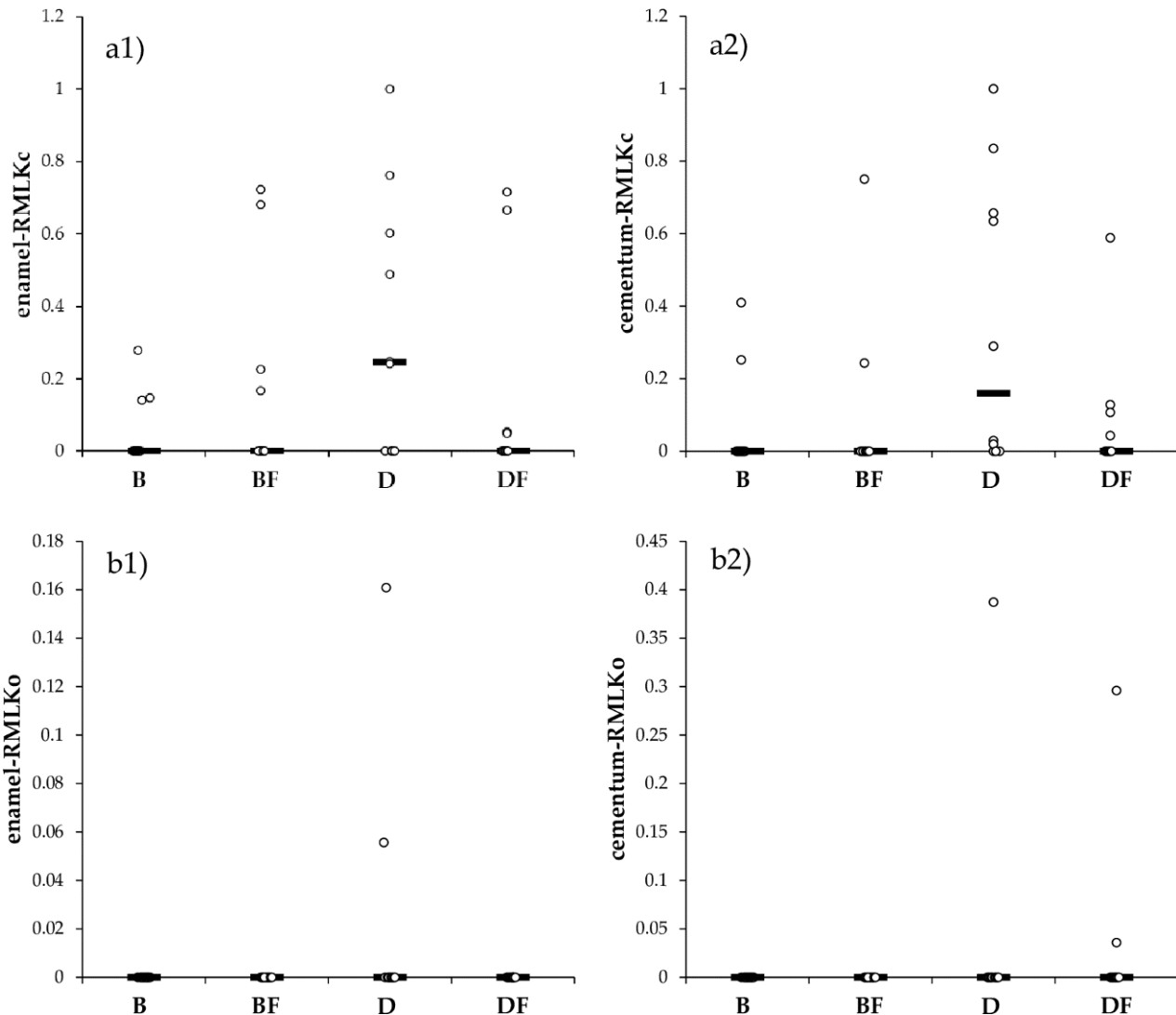

**Figure 2.** Variability of RMLK by type of restoration and material: RMLKc for restorations located in enamel (**a1**) and cementum (**a2**). RMLKo for restorations located in enamel (**b1**) and cementum (**b2**). RMLK = ratio of marginal microleakage; RMLKc = cervical RMLK; RMLKo = occlusal RMLK; B = Beautiful II LS®; BF = Beautiful Flow Plus®; D = Dynamic Plus®; DF = Dynamic Flow®.

From a qualitative point of view, marginal microleakage was identified as a pink line of different dimensions infiltrating the tooth–resin composite interface. Generally, colored infiltrations around restorations were seldom found on tooth sections (Figures 3c, 4a and 5a) or on optical microscopy images (Figure 5b) regardless of the type of material. In the majority of tooth sections, no infiltration around restorations was observed (Figures 3e and 6a).

In all the cases of composite resins included in our testing procedure, for both types of restorations, the optical microscopy images showed a particular degree of marginal microleakage (Figure 5b) emerging in the cervical part of the restorations. However, for most of the sections, no dye penetration was identified (Figures 4b and 6c). Very seldom microleakage was observed by optical microscopy into occlusal part of the restorations. Non-cavitated occlusal caries were additionally identified (Figure S1d).

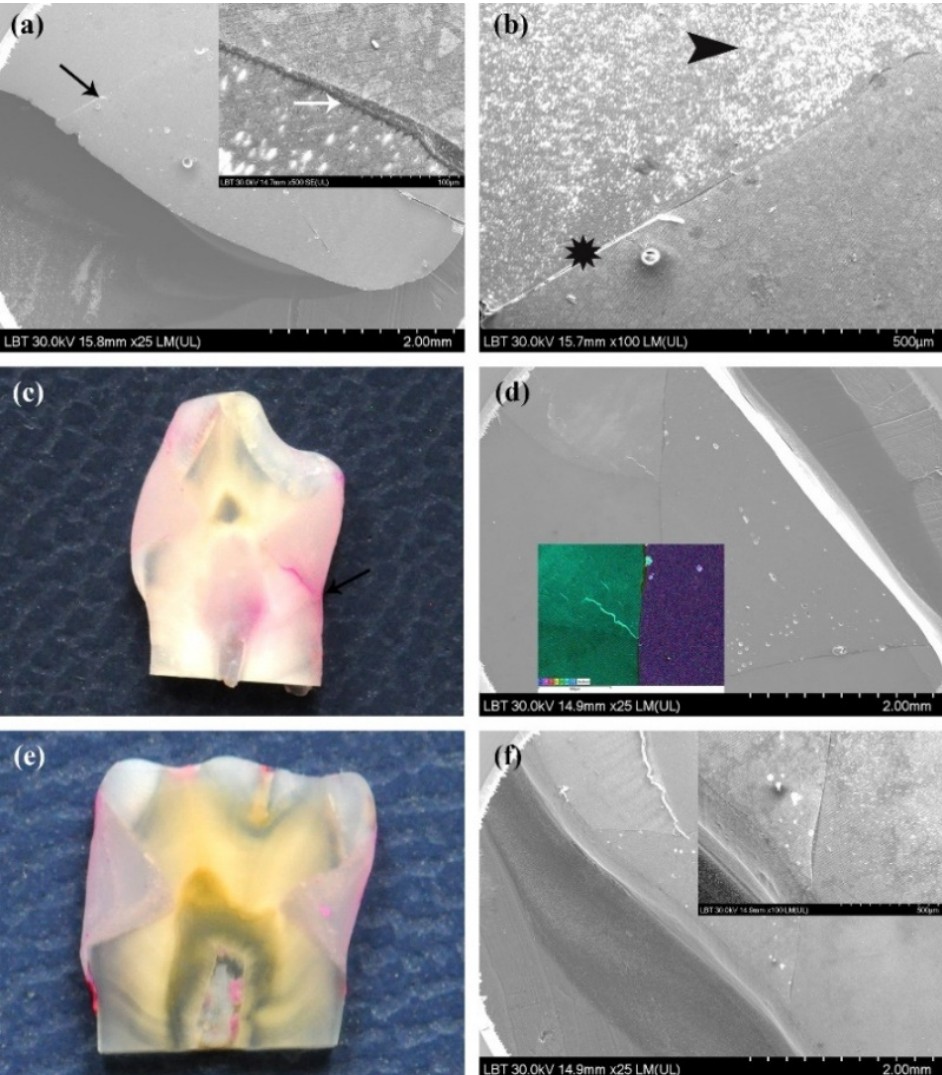

**Figure 3.** Beautiful II LS® restorations. Cervical microleakage of a cementum-located restoration (25×) continuous with a dark-colored intact adhesive layer. Inset, resinous rete tags of about 5 μm (500×) (**a**). Detail of the cervical cracked adhesive layer; dentin tubuli almost perpendicular to the surface (upper-left) (100×) (**b**). Infiltrated cementum-located restoration (right) corresponding to picture (a) and well-adapted enamel-located restoration (left) corresponding to picture d (**c**). Good cervical adhesion of an enamel-located restoration (lower right) and cracked occlusal enamel (upper left) not affecting the adhesive interface (25×). Inset, detailed fissures in occlusal enamel, green-colored dentin due to increased Ca, O, and P content, and purple-colored restoration due to Si, Al, and Sr content (50×) (**d**). Good cervical adhesion of a cementum-located restoration on tooth section (right) (**e**) and SEM (25×). Inset, detailed cervical intact adhesive layer (100×) (**f**). Black arrow, asterisk = microleakage; white arrow = resin tags, arrowhead = dentin tubuli.

*3.2. Scanning Electron Microscopy Results*

SEM analysis confirmed the results highlighted by optical microscopy investigation: a good overall peripheral adhesion of the restorations (Figures 3d,f and 6b) and occasional infiltrations mostly in the cervical areas of the restorations (Figures 3a and 5c) but also randomly located at the level of adhesive interface (Figure 4c,d). An intact adhesive layer in the occlusal part of the restorations was mostly found (Figures 3d,f, 4c, 5c and 6b,d). However, fissures in occlusal enamel without leakage of the adhesive interface were observed (Figures 3d,f, 4c, 5c,d and 6b,d).

For the majority of restorations, a relative continuous adhesive layer was observed. This thin layer revealed good homogeneity and was composed of uniformly thick ($\approx$2 µm) (Figure 3f) electron-dense structures (Figures 3d,f, 5c, and 6b) that virtually conformed to the tooth surface. Areas displaying increased thickness of the adhesive layer ranging between 2 and 10 µm were also found (Figures 4f, 5d,e, and 6d). Well-formed resinous tags penetrating into the dentin ranging from 5 to 10 µm (Figure 3a inset, Figures 5f and 6b) depending on the angulation of the section related to the tooth surface were observed. Normal dentin structure with dentin tubuli of about 2 µm diameter (Figures 3a,b and 4d,e), and mosaic-like aspects of the restorations associated with filler load (Figures 3a,b, 4f, 5e and 6b) could also been described.

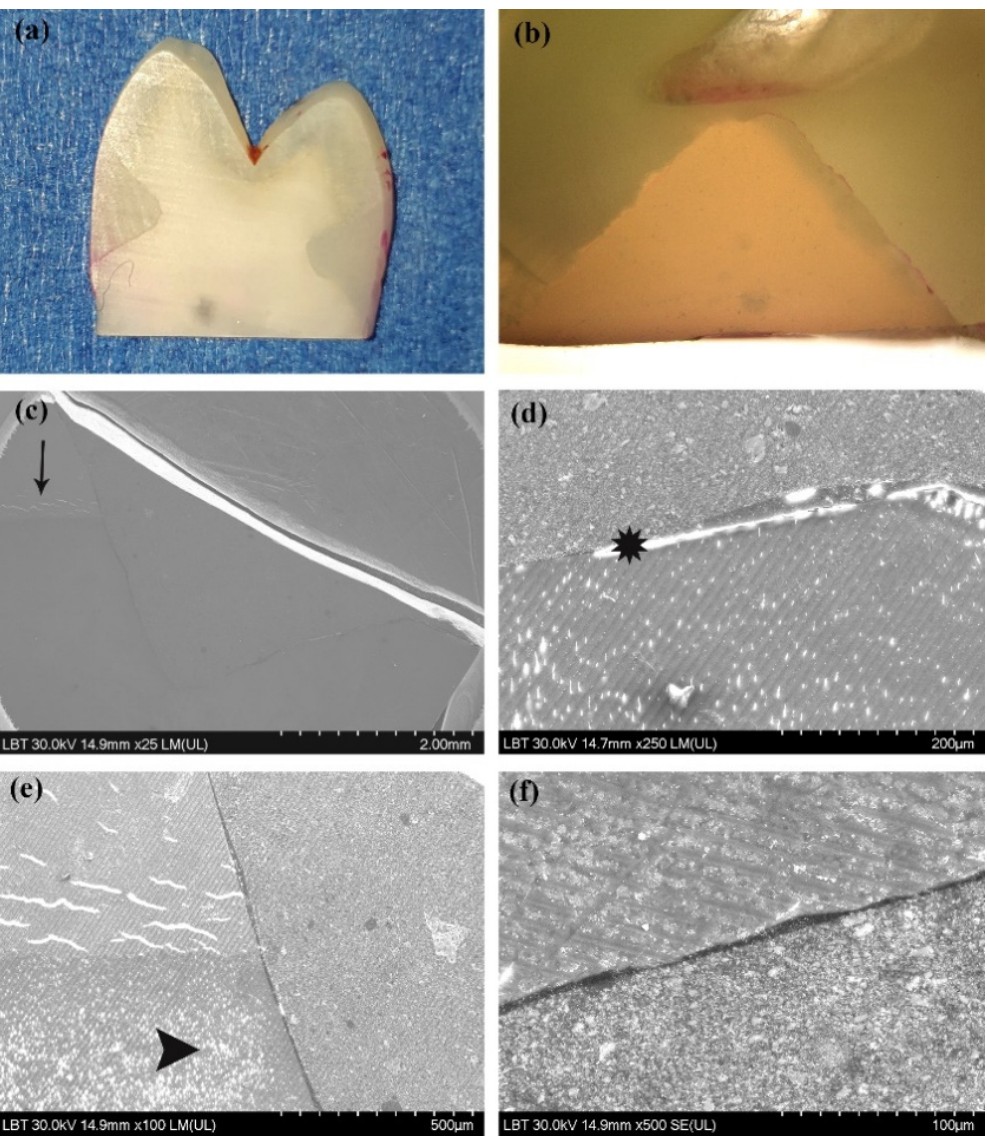

**Figure 4.** Beautiful Flow II Plus® restoration. Good cervical adhesion of the cementum-located restoration on the tooth section (right) (**a**) and on optical microscopy (**b**). SEM overview of the same restoration with good cervical adhesion and some white fissures in the occlusal enamel (upper-left) (25×) (**c**). Random microleakage of the adhesive layer (250×) (**d**). Detailed fissures in occlusal enamel (upper left) (100×) (**e**). Continuous 5 µm thick adhesive layer (500×) (**f**). black arrow = enamel fissures, asterisk = microleakage, arrowhead = dentin tubuli.

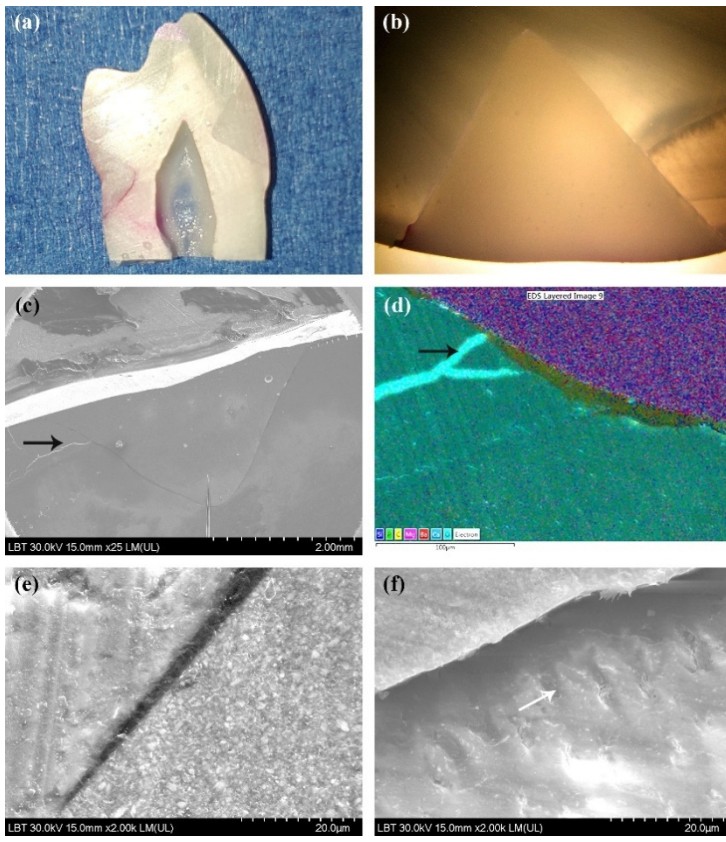

**Figure 5.** Dynamic Plus® restoration. Infiltrated cervical area of the cementum-located restoration on the tooth section (left) (**a**), optical microscopy (**b**), and SEM image (upper-right) (25×) (**c**). Cracked occlusal enamel on energy-dispersive X-ray analysis (EDX) image (500×) (**d**) not interfering the adhesive layer of 2–6 μm (2000×) (**e**). Well-designed resinous rete-tags of about 2 μm and adhesive layer of 6–8 μm (2000×) (**f**). Black arrow = cracked enamel; white arrow = resin tag.

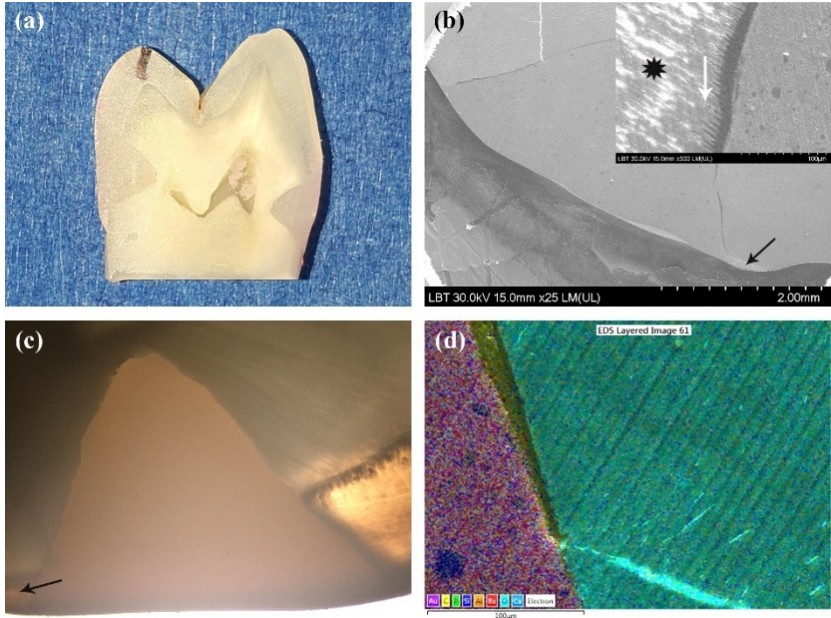

**Figure 6.** Dynamic Flow® restorations. Good adhesion (**a**). Intact adhesive interface of the enamel-located restoration on SEM image (20×). Inset, perfect visible 20 μm long rete tags (500×) (**b**). Optical microscopy of restoration b (**c**). Cracked cervical enamel on EDX image not interfering with the adhesive layer (500×) (**d**). black arrow = cervical enamel; white arrow = resin tags; asterisk = dentin tubuli.

*3.3. Energy-Dispersive X-ray Analysis*

The prepared samples containing tooth–restoration structures were assessed using EDX analysis. For the purpose of this study, 10 sections for each restorative material were evaluated. Three specific regions of interest (ROIs) were defined for each of the selected samples (resin composite, enamel, and dentine), and the element content values were provided as percentage weight (wt %). Mean values (± SD) for the results are included in Table S1. As predicted, it could be observed that the composition of tested dental materials was not uniform. One example included silicon (Si). This type of content was higher in conventional composites than in correspondent flow composites (B vs. BF = 9.2 ± 1.09 vs. 4.9 ± 0.1 and D vs. DF = 15.68 ± 1.10 vs. 11.66 ± 2.04) due to the higher glass-based filler load in conventional composites. However, it could be found that all resin samples displayed a predominance of oxygen (O), carbon (C), and silicon elements. Contrastingly, no trace of strontium (Sr) could be identified in D or DF, whereas similarly high levels could be detected in B and BF (8.92 ± 0.58 and 8.56 ± 0.61). On the contrary, barium (Ba) was detected in high levels in D and DF materials (11.14 ± 1.27 and 11.14 ± 3.15), but it was almost non-detected in the other two composites. Low levels of fluoride (F) were detected only in B and BF materials. The concentration of oxygen, carbon, calcium, and phosphorus (P) were relatively similar for enamel and dentin specimens.

## 4. Discussion

It has been shown that the study of the adhesion mechanisms of resin-based materials is highly dependent on many variables related to material composition, dental substrates, tooth lesion morphology, or treatment protocol. The restoration of NCCLs is still challenging in clinical practice [3,34] due to the complex dental structural modifications of affected surfaces and the proximity of the gingival margin [3]. Moreover, the insufficient bondable enamel in cervical area increases the susceptibility to microleakage and recurrent decay [35]. In NCCL restorations, polymerization shrinkage of resin composites increases the stress on the cervical adhesive interface, possibly developing a high risk of bonding failure. In addition, this process may also result in dental as well as periodontally relevant complications [35,36]. The literature recommended further research to identify materials and techniques providing the optimal results in NCCL restorations.

On account of the microleakage testing conducted in the present study, it can be concluded that a certain amount of microleakage was recorded for all tested materials, which is mostly located in the cervical part of the restorations. It is important to underline that other similar in vitro studies [37] reported analogous results. Most of the samples displayed no microleakage. However, microleakage associated with resin composites is an expected phenomenon [38] and could not be avoided even under ideal laboratory conditions [26,33]. Results concerning the adhesive capabilities of restorative materials failed to reject our null hypothesis, as there was no statistically significant variation in adhesive efficiency. The evaluation of marginal microleakage recorded several ratio differences. Our interpretation of the results is that such ratio differences of microleakage emerged from random variation. Our data are also in agreement with the results of other microleakage studies using flowable composites [39]. However, Fruits et al. showed better results of in vitro flowable composite cervical restorations in comparison with conventional composites [40].

The good adhesion recorded for tested materials recommends both types of resin composites in clinical practice to restore NCCLs. The set of four materials chosen in the present study represents an element of novelty. Moreover, to our knowledge, there are no in vitro adhesion studies on such lesions using the low shrinkage BF material. The use of conventional and flow materials from the same manufacturer completes the novelty of the present investigation.

Flowable resin composites have been increasingly implemented in practice as an alternative to conventional composites in many clinical situations including the restoration of NCCLs [41] due to their good polishability, relative performant optical properties, biocompatibility, adhesive properties, and reduced low elastic modulus [17]. The low elastic

modulus of flowable composites in comparison with conventional hybrid composites could compensate for the increased polymerization shrinkage stress [42] prevent the failure of the marginal adhesive seal in cervical restorations [43], which could explain the results of our study. The D material has the lowest filler content and a low elastic modulus, but its adhesive behavior was comparable with those of the other materials. The high wettability of the dental surfaces ensuring an intimate contact with the adhesive layer is an advantage of flowable composites in comparison with conventional ones [17]. However, many clinical details related to the manoeuvrability of both flowable and conventional composites, materials' application steps, curing protocol, as well as performance of the isolation field highly influence the restoration fate beyond the composition of dental materials [8,14,17]. Controlling moisture during restoration influences the prognosis of class V restorations, but the optimal isolation of the operative field still remains a difficult task [16].

The SEM micrographs and optical microscopy images of most sections revealed well-adapted, continuous adhesive interfaces underneath the restorations. Occasionally, disruption of the adhesive interface was found. These observations are sustained by others [26]. In our study, one-step self-etch adhesives were used as recommended by the manufacturers, although they probably are not the best option for restoring NCCLs [6,44]. The literature data report that the combination of primer and adhesive in the same bottle tends to decrease the bonding strength, thereby influencing the hybrid layer durability [45]. As a result, the survival of the restorations is affected. However, both adhesives used in the present study are classified as mild acidic adhesive systems associated with an increased bonding efficiency in comparison with the more acidic adhesives from the same class [6].

In the present study, well-formed more electron-dense structures of 5 to 10 μm penetrating into the dentin and resembling to resinous tags were observed for both adhesives. Usually, mild self-etch adhesives demineralize dentin only very shallowly and remove incompletely the smear layer and smear plugs from the dentine tubuli, providing a submicronic hybrid layer and resinous tags of about 2 μm [46]. The increased dimensions of resin tags observed by the present study may be related to the increased demineralization performance of current self-etch adhesives and to other influencing factors, such as agitation during application, thickness of the smear layer, and viscosity and wetting characteristics of the products [47]. However, it seems that that the thickness of the hybrid layer and the presence of resin tags do not significantly influence the bonding performance of the adhesives [48].

In this study, the cervical locations of the cavity margins did not influence the adhesive performances of the materials. No differences were highlighted between cervical and occlusal microleakage ratios as well. The data in the literature on this issue are inconsistent [37,49]. An increased cervical leakage may be due to the irregular prismatic structure of the cervical enamel [50] or to the presence of cervical cementum [51] that induces the formation of rare resin tags and thus a weaker bonding area [51].

One limitation of this study is related to the investigated sample size. The variability of the measurements is very high, and thus, small differences between groups could not be detected. The investigation on a larger number of repetitions must be conducted to validate the results of the present study.

Although conventional classical class V cavities are usually designed to test composite adhesion [43], the present study prepared wedge-shaped cavities based on the clinical reality that reports increasingly common such clinical situations [52]. The dual localization of the lesions at the cervical level, above or beyond the enamel–cement junction, resulted from the still unperfected clinical problem of the uncertain adhesion in the cervical area [50,51]. For better observing the materials' adhesive behavior at cervical margins, the present study provided constant high bond strength at the occlusal part of the restorations by preparing an enamel bevel [53] and selective enamel etching with 35% phosphoric acid [6].

Systematic standardization was performed so as to minimize variability in this experiment. Consequently, the authors were committed to a very precise and detailed restoration technique for obtaining a minimal polymerization shrinkage and better marginal adapta-

tion [8,54]. Thorough analysis of the restorative steps as derived from clinical recommendations associated with the preparation of a specific form of cervical defects constitutes an element of originality of our study.

A limitation of the present study was the impossibility to provide a profound dentin substrate resembling the NCCLs represented by a heterogeneous hypermineralized dentin layer [55,56], obliteration of the majority of dentinal tubules [26] and, eventually, an altered composition and microstructure of the enamel and dentin consecutive to the aging process [3]. It is known that the substrate structure has an important role in the adhesion of directly placed composite restorations [57,58].

A semi-quantitative elemental composition for the three components of sections was obtained by EDX analysis. Sr was detected only in B and BF materials, and Ba was detected only in D and DF materials. Barium from barium oxide glass fillers was employed for providing higher radiopacity [59]. Barium and strontium glasses were required to adjust the refractive index of the filler particles and to increase the transparency of the mixture. Fluoride content in B and BF materials could improve their clinical behavior [19]. Some differences between conventional and the correspondent flowable composites have been observed due to the increased filler load in conventional composites. The results of our EDX analyses confirm the qualitative compositional data claimed by the manufacturers. Our results show that there was a minimal variability in enamel and dentin compositions, respectively, which could be partially responsible for the adhesive behavior of composite resins [60].

## 5. Conclusions

It is difficult to avoid the microleakage of NCCL restorations. The enamel-located and cementum-located cervical margins of the restorations did not influence the adhesive performances of the materials. No differences were observed between cervical and occlusal microleakage ratios as well. Based on the present results, both conventional and flowable resin composites are recommended in the clinic to restore NCCLs. EDX evaluations show a minimal variability of dental substrate, which could explain in part the similar adhesive behavior of the restorations. EDX sustains the qualitative compositions as provided by the manufacturers.

**Supplementary Materials:** The following are available online at https://www.mdpi.com/article/10.3390/app11073173/s1, Figure S1: Method details. Prepared cavities on experimental teeth (a). Silicon mounting template for an experimental group of teeth (b). An experimental group of teeth embedded in autopolymerising acrylic resin (c). Incipient dentin decay (d). Table S1: Energy-dispersive X-ray analysis of elemental composition in the tested composite resin materials [percent weight (wt%)].

**Author Contributions:** Conceptualization, A.G.D., A.R., P.Ș. and A.S.; Data curation, S.D.B.; Formal analysis, S.D.B.; Funding acquisition, D.B.O.; Investigation, D.B.O., L.B.T., A.G.D., A.C., C.G. and A.S.; Methodology, A.G.D., A.M. and A.S.; Project administration, A.S.; Resources, D.B.O. and A.C.; Writing—original draft, A.C. and A.R.; Writing—review and editing, P.Ș., D.B.O., L.B.T., C.G. and A.M. All authors have read and agreed to the published version of the manuscript.

**Funding:** This research was funded by Iuliu Hațieganu University of Medicine and Pharmacy Cluj-Napoca, PCD grant number 2462/8/17.01.2020.

**Institutional Review Board Statement:** The study was conducted according to the guidelines of the Declaration of Helsinki, and approved by the Ethical Board of the Iuliu Hațieganu University of Medicine and Pharmacy Cluj-Napoca (No. 268/30.07.2019).

**Informed Consent Statement:** Informed consent was obtained from all subjects involved in the study to use the extracted teeth for research purposes.

**Data Availability Statement:** The datasets generated during the current study are available from the corresponding author on reasonable request.

**Acknowledgments:** We are grateful to Adrian Florea for his assistance in preparing the illustrations.

**Conflicts of Interest:** The authors declare no conflict of interest.

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
