# Peer review of "Adhesion of Flowable Resin Composites in Simulated Wedge-Shaped Cervical Lesions: An In Vitro Pilot Study"

_applsci, doi:10.3390/app11073173_

Round 1

Reviewer 1 Report

In this paper the evaluation of adhesion of two flowable and two conventional hybrid resin composites in simulated  wedge-shaped cervical lesions was performed by an in vitro pilot study with significant results. This manuscript present novelty and originality for scientific world and worth publishing. 
However, it requires some minor technical corrections in accordance with with author guidelines:
1.    The tables should be presented with horizontal lines and along the entire width of the page. The abbreviations should be placed at the footer of the tables. 
2.    Figures captions should be placed below to the Figures. In figure captions of the Figure 4, “of the” is doubled, in the sentence 2 “SEM overview….(c)” and must be deleted.

Author Response

Dear Sir,

We would want to thank you for your comments that helped us to valorise our work.

Please find a point-by-point response to the issues raised related to the minor revision of the “applsci-1161196” article and the corresponding modification of the manuscript traced by “Track changes”.

Thank you for your time,

Alexandra Roman, corresponding author

In this paper the evaluation of adhesion of two flowable and two conventional hybrid resin composites in simulated  wedge-shaped cervical lesions was performed by an in vitro pilot study with significant results. This manuscript present novelty and originality for scientific world and worth publishing. 
However, it requires some minor technical corrections in accordance with with author guidelines:
1.    The tables should be presented with horizontal lines and along the entire width of the page. The abbreviations should be placed at the footer of the tables. 

Answer: Thank you for this observation. We performed the mentioned correction: the vertical lines were eliminated, and the abbreviations were placed at the footer of the tables.

  1.   Figures captions should be placed below to the Figures. In figure captions of the Figure 4, “of the” is doubled, in the sentence 2 “SEM overview….(c)” and must be deleted.

Answer: The captions were placed bellow the figures, the double expression was erased. We also repositioned the pictures in order to fit better on the page.

Reviewer 2 Report

Dear Authors, it is very interesting article about different methods of NCCL treatment. It is evident that you have put a lot of work into planning and carrying out the research. I see only a few minors errors.

abstract line 40: "These composites" may be confusing to readers which material is involved. please specify.

material and method line 174: which mode of polymerization lamp was used? Different modes can influence on material shrinkage and hence marginal integrity.

material nad method line 176: How the samples was polished?

discussion lines 395-398: This paragraph seems like an interruption. It does not connect to the previous or next.

discussion line 452-459: this paragraph seems like a description from  material and method section not from discussion section.

Conclusion line 486-489: it looks like a results not a conclusion. 

Author Response

Dear Sir,

We would want to thank you for your comments that helped us to valorise our work.

Please find a point-by-point response to the issues raised related to the minor revision of the “applsci-1161196” article and the corresponding modification of the manuscript.

Thank you for your time,

Alexandra Roman, corresponding author

Dear Authors, it is very interesting article about different methods of NCCL treatment. It is evident that you have put a lot of work into planning and carrying out the research. I see only a few minors errors.

1.abstract line 40: "These composites" may be confusing to readers which material is involved. please specify.

Answer: “These composites” was replaced with “The four experimental composites”. In order to maintain the maximum 200 words of the abstract, a sentence was shortened, please find the modifications in manuscript.

2.material and method line 174: which mode of polymerization lamp was used? Different modes can influence on material shrinkage and hence marginal integrity.

Answer: Thank you for this observation. We added in text that it was a conventional polymerization mode.

3.material nad method line 176: How the samples was polished?

Answer: The samples were polished with medium and fine polishing discs (OptidiscTM, Kerr Corporation). Please find this information added in the Material and Methods.

4.discussion lines 395-398: This paragraph seems like an interruption. It does not connect to the previous or next.

Answer: Thank you for this observation. We erased this paragraph.

5.discussion line 452-459: this paragraph seems like a description from material and method section not from discussion section.

Answer: This paragraph explains the background that sustains the choice of our restorative technique. We rephrased the paragraph in order to fit better to the Discussion section. Please find the modified paragraph in the main document. The final version of the paragraph is as follows: “Systematic standardization was performed so as to minimize variability in this experiment. Consequently, the authors were committed to a very precise and detailed restoration technique for obtaining a minimal polymerization shrinkage and better marginal adaptation [8,54]. Thorough respect of restorative steps as derived from clinical recommendations associated with the preparation of a specific form of cervical defects constitute an element of originality of our study”.

6.Conclusion line 486-489: it looks like a result not a conclusion. 

Reviewer 3 Report

The article: " Adhesion of flowable resin composites in simulated 3 wedge-shaped cervical lesions: an in-vitro pilot study" investigated the adhesive properties of flowable resin composites placed to restore non-carious cervical lesions compared with conventional hybrid composite resins. The article presents a good structure. The Material and Methods are well described and the authors used the correct methodology to assess this topic. However, there are some corrections to follow before a definitive publication.

Line 101: Adds other references regarding the application of SEM in dentistry to explain the use of this tool in dentistry research.

For example :

Gayatri C, Rambabu T, Sajjan G, Battina P, Priyadarshini MS, Sowjanya BL. Evaluation of Marginal Adaptation of a Self-Adhering Flowable Composite Resin Liner: A Scanning Electron Microscopic Study. Contemp Clin Dent. 2018 Sep;9(Suppl 2): S240-S245

Di Fiore A, Mazzoleni S, Fantin F, Favero L, De Francesco M, Stellini E. Evaluation of three different manual techniques of sharpening curettes through a scanning electron microscope: a randomized controlled experimental study. Int J Dent Hyg. 2015 May;13(2):145-50. 

Line 106:  The authors could add new references regarding the fields of engineering and chemistry.  

Line 118: How did you conserve the tooth? Explain the method.

Discussion Chapter.

The author may synthesize the discussion. It is long-winded.

Author Response

Dear Sir,

We would want to thank you for your comments that helped us to valorise our work.

Please find a point-by-point response to the issues raised related to the minor revision of the “applsci-1161196” article and the corresponding modification of the manuscript.

Thank you for your time,

Alexandra Roman, corresponding author

The article: “ Adhesion of flowable resin composites in simulated 3 wedge-shaped cervical lesions: an in-vitro pilot study” investigated the adhesive properties of flowable resin composites placed to restore non-carious cervical lesions compared with conventional hybrid composite resins. The article presents a good structure. The Material and Methods are well described and the authors used the correct methodology to assess this topic. However, there are some corrections to follow before a definitive publication.

1.Line 101: Adds other references regarding the application of SEM in dentistry to explain the use of this tool in dentistry research.

For example :

Gayatri C, Rambabu T, Sajjan G, Battina P, Priyadarshini MS, Sowjanya BL. Evaluation of Marginal Adaptation of a Self-Adhering Flowable Composite Resin Liner: A Scanning Electron Microscopic Study. Contemp Clin Dent. 2018 Sep;9(Suppl 2): S240-S245

Di Fiore A, Mazzoleni S, Fantin F, Favero L, De Francesco M, Stellini E. Evaluation of three different manual techniques of sharpening curettes through a scanning electron microscope: a randomized controlled experimental study. Int J Dent Hyg. 2015 May;13(2):145-50. 

Answer: We added the above-mentioned references in text as requested as well as in the reference list.

2.Line 106:  The authors could add new references regarding the fields of engineering and chemistry.  

Answer: We added two new references, please find them in text and in the reference list.

-Abdolahpur Monikh F, Chupani L, Vijver MG, Vancová M, Peijnenburg WJGM. Analytical approaches for characterizing and quantifying engineered nanoparticles in biological matrices from an (eco)toxicological perspective: old challenges, new methods and techniques. Sci Total Environ. 2019 Apr 10;660:1283-1293. doi: 10.1016/j.scitotenv.2019.01.105

-Son D, Cho S, Nam J, Lee H, Kim M. X-ray-Based Spectroscopic Techniques for characterization of Polymer Nanocomposite Materials at a Molecular Level.  Polymers (Basel). 2020 May 4;12(5):1053. doi: 10.3390/polym12051053

3.Line 118: How did you conserve the tooth? Explain the method.

Answer: Here is only a brief description of the research phases (study design). The information on tooth storage was already mentioned in row 156-159.

4.Discussion Chapter.

The author may synthesize the discussion. It is long-winded.

Answer: Thank you for this suggestion. We erased a few paragraphs and shortened some sentences, Please find in the main document these modifications.
